# Graph Representation Learning via Causal Diffusion for Out-of-Distribution Recommendation

## Abstract

Graph Neural Networks (GNNs)-based recommendation algorithms typically assume that training and testing data are drawn from independent and identically distributed (IID) spaces. However, this assumption often fails in the presence of out-of-distribution (OOD) data, resulting in significant performance degradation. In this study, we construct a Structural Causal Model (SCM) to analyze interaction data, revealing that environmental confounders (e.g., the COVID-19 pandemic) lead to unstable correlations in GNN-based models, thus impairing their generalization to OOD data. To address this issue, we propose a novel approach, graph representation learning via causal diffusion (**CausalDiffRec**) for OOD recommendation. This method enhances the model's generalization on OOD data by eliminating environmental confounding factors and learning invariant graph representations. Specifically, we use backdoor adjustment and variational inference to infer the real environmental distribution, thereby eliminating the impact of environmental confounders. This inferred distribution is then used as prior knowledge to guide the representation learning in the reverse phase of the diffusion process to learn the invariant representation. In addition, we provide a theoretical derivation that proves optimizing the objective function of CausalDiffRec can encourage the model to learn environment-invariant graph representations, thereby achieving excellent generalization performance in recommendations under distribution shifts. Our extensive experiments validate the effectiveness of CausalDiffRec in improving the generalization of OOD data, and the average improvement is up to 10.69% on Food, 18.83% on KuaiRec, 22.41% on Yelp2018, and 11.65% on Douban datasets. Our code is available at https://anonymous.4open.science/r/CausalDiffRec-33D1.

## CCS Concepts

• **Information systems** → **Recommender systems**.

## Keywords

Graph Recommendation, Distributionally Robust Optimization, Out-of-Distribution

**ACM Reference Format:**
Anonymous Author(s). 2018. Graph Representation Learning via Causal Diffusion for Out-of-Distribution Recommendation. In *Proceedings of Make sure to enter the correct conference title from your rights confirmation emai (Conference acronym 'XX).* ACM, New York, NY, USA, 13 pages. https://doi.org/XXXXXXX.XXXXXXX

## 1 Introduction

Graph Neural Networks (GNN) [43, 57, 58], due to their exceptional ability to learn high-order features, have been widely applied in recommendation systems. GNN-based recommendation algorithms [8, 30, 42] learn user and item representations by aggregating information from neighboring nodes in the user-item interaction graph and then computing their similarity to predict user preferences. In addition, researchers have introduced various other techniques to continuously improve GNN-based recommendation algorithms. For example, integrating attention mechanisms [7, 25] with knowledge graphs [14] led to improving recommendation accuracy. Furthermore, the introduction of contrastive learning aims to improve the robustness of recommendation algorithms [17, 17, 50].

Despite significant progress in enhancing recommendation accuracy, most existing methods assume that test and training datasets follow an independently and identically distributed (IID) pattern, focusing on performance improvements under this assumption. Such methods struggle to generalize effectively to out-of-distribution (OOD) data, where test data distributions markedly differ from training data [13, 23, 47]. For example, as shown in Figure 1, the popularity of medical supplies surged during the COVID-19 pandemic, alongside increased demand for fitness equipment and electronics due to government-imposed homestays. The recommender systems might infer that purchasing masks correlates with buying these additional items, driven by the common 'pandemic' factor rather than direct causation. Once the pandemic subsides, demand shifts, reducing the popularity of masks and related items. Consequently, the system may inaccurately recommend fitness equipment and electronics to mask buyers under the new distribution, leading to poor performance. Additionally, tests on the Yelp2018 dataset, comparing IID and OOD sets, demonstrate a significant performance drop. LightGCN [9] experiences an average 29.03% decline across three metrics on OOD data compared to IID settings, highlighting the robustness issues of GNN-based models in OOD scenarios. This challenge motivates the development of a recommendation framework with strong generalization capabilities for handling distribution shifts.

Several studies have aimed to enhance the generalization of recommender systems on OOD datasets by using causal inference to address data distribution shifts. For instance, CausPref [11] builds on NeuMF [10] by implementing invariant user preference causal learning and anti-preference negative sampling to boost model generalization. COR [31] utilizes a Variational Auto-Encoder for causal modeling by inferring unobserved user features from historical interactions. However, these methods are not tailored for GNNs, complicating their adaptation to GNN-based approaches.

Other researchers have adopted techniques like graph contrastive learning and graph data augmentation to improve the robustness of GNN-based recommendation algorithms, such as SGL [40], SimGCL [49], and LightGCL [1]. These approaches mainly address noise or popularity bias but underperform when test data distributions are

unknown or varied, as evidenced by experimental results. Recently, a few GNN-based methods [52] have been proposed to enhance generalization across multiple distributions, but they lack strong theoretical backing.

Given these limitations, there is an urgent need to design theoretically grounded GNN-based methods to address distribution shifts. In this paper, we use invariant learning to improve the generalization of the OOD dataset. Utilizing insights from the prior knowledge of environment distribution and invariant learning [5, 20, 56] enhances model stability across varied environments. This is achieved by acquiring invariant representations, boosting the model's generalization capabilities and overall robustness. However, designing models based on invariant learning still faces the following two challenges:

- **(1)** How to infer the distribution of underlying environments from observed user-item interaction data?
- **(2)** How to recognize environment-invariant patterns amid changing user behaviors and preferences?

In this paper, we first develop a Structural Causal Model (SCM) to analyze data generation processes in recommender systems, specifically addressing the impact of data distribution shifts on GNN-based recommendation algorithms. We find that latent environmental variables can lead these models to capture unstable correlations, hindering their generalization to OOD data. To tackle this, we introduce CausalDiffRec, a novel method using causal inference to remove these unstable correlations by learning invariant representations across different environments. CausalDiffRec comprises an environment generator to create diverse data distributions, an environment inference module to identify and utilize environmental components, and a diffusion module guiding invariant representation learning. Theoretically, we prove that CausalDiffRec can achieve better OOD generalization by identifying invariant representations across varying environments.

The contributions of this paper are concluded as follows:

- **Causal Analysis**. We construct the SCM and analyze the generalization ability of GNN-based recommendation models on OOD data from the perspective of data generation. Based on our analysis and experimental results, we conclude that environmental confounders lead the model to capture unstable correlations, which is the key reason for its failure to generalize under distribution shifts.
- **Methodology**. We introduce CausalDiffRec, a new GNN-based method for OOD recommendation, comprising three main modules: environment generation, environment inference, and diffusion. The environment generation module simulates user data distributions under various conditions, the environment inference module uses causal inference and variational approximation to deduce environment distribution, and the diffusion module facilitates graph representation learning. Theoretical analysis confirms that optimizing CausalDiffRec's objective function enhances model generalization.
- **Experimental Findings**. We constructed three common types of distribution shifts across four datasets and conducted comparative experiments. The experiments demonstrate that CausalDiffRec consistently outperforms baseline methods. Specifically, CausalDiffRec exhibits enhanced generalization capabilities when

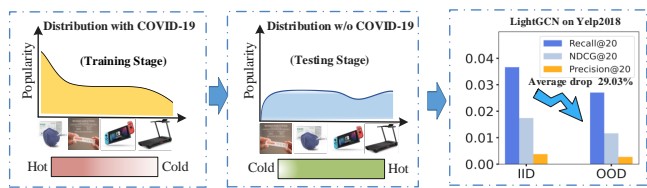

**Figure 1: Left and Middle: An example illustrates the popularity distribution shift, i.e., how the popularity of masks, disinfectants, exercise equipment, and electronic products changes with the COVID-19 pandemic. Right: We constructed both IID and OOD sets on the Yelp2018 dataset and compared the performance of the LightGCN model [9] on these datasets. We found a significant average performance drop (i.e., 29.03%) in OOD data across three metrics.**

dealing with OOD data, achieving a maximum metric improvement rate of 36.73%.

## 2 Preliminary

### 2.1 GNN-based Recommendation

Given the observed implicit interaction matrix $\mathcal{R} \in \{0, 1\}^{m \times n}$, in which $\mathcal{U} = \{u_1, u_2, \ldots, u_m\}$ represents the set of users, $\mathcal{I} = \{i_1, i_2, \ldots, i_n\}$ represents the set of items, $m$ and $n$ denote the number of users and items, respectively. For the elements in the interaction matrix, $r_{ui} = 1$ indicates an interaction between user $u$ and item $i$, otherwise 0. In GNN-based recommendation algorithms, the user-item interaction matrix $\mathcal{R}$ is first transformed into a bipartite graph $G = \{\mathcal{V}, \mathcal{E}\}$. We employ $\mathcal{V}$ to represent the node set and $\mathcal{E} = \{(u, i) | u \in \mathcal{U}, i \in \mathcal{I}, r_{ui} = 1\}$ denotes the edge set. Given a user-item interaction graph $\mathcal{G}_u$ and the true user interactions $y_u$ with respect to (w.r.t) user $u$, the optimization objective of GNN-based methods can be expressed as:

$$\arg \min_\theta \mathbb{E}_{(\mathcal{G}_u, y_u) \sim P(G, Y)}[l(f_\theta(\mathcal{G}_u; \theta), y_u)], \quad (1)$$

where $f_\theta(\cdot)$ is a learner that learns representations by aggregating high-order neighbor information from the user-item interaction graph. $l$ denotes the loss function and $P(G, Y)$ represents the joint distribution of the interaction graph $G$ and true label $Y$.

### 2.2 Denoising Diffusion Probabilistic Models

Denoising Diffusion Probabilistic Models (DDPM) [12] have been widely used in the field of image and video generation. The key idea of DDPM is to achieve the generation and reconstruction of the input data distribution through a process of gradually adding and removing noise. It leverages neural networks to learn the reverse denoising process from noise to real data distribution.

The diffusion process in recommender system models the evolution of user preferences and item information through noise addition and iterative recovery. Initially, data $\mathbf{x}_0$ sampled from $q(\mathbf{x})$ undergo a forward diffusion to generate noisy samples $\mathbf{x}_1, \ldots, \mathbf{x}_T$ over $T$ steps. Each step adds Gaussian noise, transforming the data distribution incrementally [12]:

$$q(\mathbf{x}_t | \mathbf{x}_{t-1}) = \mathcal{N}\left(\mathbf{x}_t; \sqrt{1 - \beta_t} \mathbf{x}_{t-1}, \beta_t \mathbf{I}\right), \quad (2)$$

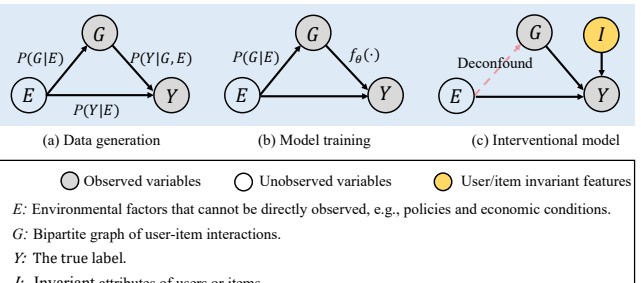

(a) Data generation     (b) Model training     (c) Interventional model

⬤ Observed variables    ◯ Unobserved variables    🟡 User/item invariant features

$E$: Environmental factors that cannot be directly observed, e.g., policies and economic conditions.

$G$: Bipartite graph of user-item interactions.

$Y$: The true label.

$I$: Invariant attributes of users or items.

**Figure 2: The structure causal model for GNN-based recommendation**

where $\beta_t \in (0, 1)$ controls the level of the added noise at step $t$. In the reverse phase, the aim is to restore the original data by learning a model $p_\theta$ to approximate the reverse diffusion from $\mathbf{x}_T$ to $\mathbf{x}_0$. The process, governed by $p_\theta(\mathbf{x}_{t-1}|\mathbf{x}_t)$, uses the mean $\mu_\theta$ and covariance $\Sigma_\theta$ learned via neural networks:

$$p_\theta(\mathbf{x}_{t-1}|\mathbf{x}_t) = \mathcal{N}(\mathbf{x}_{t-1}; \mu_\theta(\mathbf{x}_t, t), \Sigma_\theta(\mathbf{x}_t, t)). \quad (3)$$

The reverse process is optimized to minimize the variational lower bound (VLB), balancing the fidelity of reconstruction and the simplicity of the model [38]:

$$\mathcal{L}_{VLB} = \mathbb{E}_{q(\mathbf{x}_{1:T}|\mathbf{x}_0)} \left[ \sum_{t=1}^{T} D_{KL}(q(\mathbf{x}_{t-1}|\mathbf{x}_t, \mathbf{x}_0)||p_\theta(\mathbf{x}_{t-1}|\mathbf{x}_t)) \right] \quad (4)$$
$$- \log p_\theta(\mathbf{x}_0|\mathbf{x}_1),$$

where $D_{KL}$ denotes the Kullback-Leibler (KL) divergence. Following [12], we mitigate the training instability issue in the model by expanding and reweighting each KL divergence term in the VLB with specific parameterization. Therefore, we have the following mean squared error loss:

$$\mathcal{L}_{\text{simple}} = \mathbb{E}_{t,\mathbf{x}_0,\boldsymbol{\epsilon}_t} \left[ \left\| \boldsymbol{\epsilon}_t - \boldsymbol{\epsilon}_\theta\left(\sqrt{\bar{\alpha}_t}\mathbf{x}_0 + \sqrt{1 - \bar{\alpha}_t}\boldsymbol{\epsilon}, t\right) \right\|^2 \right], \quad (5)$$

where $\boldsymbol{\epsilon}_t \sim \mathcal{N}(0, \mathbf{I})$ is the noise for injection in forward process, $\boldsymbol{\epsilon}_\theta(\cdot)$ denotes a function approximator that neural networks can replace, and $\alpha_t = \prod_{t=1}^{T} 1 - \beta_t$. This framework allows the model to effectively learn noise-free representations and improve recommendation accuracy.

## 2.3 Invariant Pattern Recognition Mechanism

Invariant learning [23, 51] is often used to develop predictive models that can generalize to out-of-distribution (OOD) data, which originates from different environments. Existing invariant learning methods typically rely on the following assumptions:

**Assumption**: For a given user-item interaction graph (i.e., data distribution $D$), these interaction data are collected from $K$ different environments $E$. User behavior patterns exist independently of the environment and can be used to generalize out-of-distribution user preference prediction. There exists an optimal invariant graph representation learning $F^*(\cdot)$ satisfying:

- **Invariance Property**. $\forall e \in D(E), P_\theta(Y|F^*(G), E = e, I) = P(Y|F^*(G), I)$.

- **Sufficiency Condition**. $Y = F^*(G) + \epsilon, \epsilon \perp E$, where $\perp$ indicates statistical independence and $\epsilon$ is random noise.

The invariance property assumption indicates that a graph representation learning model exists capable of learning invariant user-item representations across different data distribution environments. The sufficiency condition assumption means that the learned invariant representations enable the model to make accurate predictions.

## 3 Methodology

In this section, we first construct SCM and identify environmental confounders as the key reason for the failure of GNN-based models to generalize on OOD (out-of-distribution) data. Subsequently, we introduce the variational inference to infer the true distribution of the environment. We use the diffusion model to learn the representation based on invariant learning. Finally, we provide rigorous theoretical proof of CausalDiffRec that can achieve great generalization. The model framework is illustrated in Figure 3.

### 3.1 SCM of GNN-based Recommendation

To explore the reasons behind the failure of GNN-based models to generalize on OOD data, we follow previous works [41] [46] and first construct the SCM for data generation and data modeling in recommendation systems, as shown in Figure 2 (a) and Figure 2 (b). We find that environmental confounding factors are the key reason for the generalization failure of GNN-based methods. Finally, we design an intervention model in Figure 2 (c) to eliminate the impact of environmental confounding factors.

*3.1.1 Causal View in GNN-based Recommendation.* In Figure 2, the three causal relations are derived from the definitions of data generation. The detailed causal analysis behind them is presented as follows:

- **Environmental Factors ($E$)**: These represent unseen factors such as sudden events or policies that affect user-item interaction graphs ($G$) and true labels ($Y$). Invariant attributes ($I$), like user gender and item categories, remain unaffected by these factors. Prior research [11] suggests leveraging these invariant features can enhance model generalization in OOD environments.
- $E \rightarrow G$: This describes the direct effect of the environment on user-item interactions, defined by the probability $P(G|E)$. For instance, cold weather might increase user interactions with warm clothing.
- $G \rightarrow Y$: This reflects how the interaction graph $G$ influences the user behavior label $Y$, characterized by the GNN model $Y = f_\theta(G)$. With fixed model parameters $\theta$, the relationship between $G$ and $Y$ is deterministic.
- $I \rightarrow Y$: Invariant attributes directly affect the user behavior label $Y$. For instance, a user might consistently prefer a specific restaurant, where attributes like the location remain constant.
- $E \rightarrow Y$: The environment directly impacts the user behavior label $Y$, independent of user-item interactions. For example, during holidays, users might be more inclined to buy holiday-related items regardless of past interactions.

In real-world scenarios, training data is collected from heterogeneous environments. Therefore, the environment directly influences the distribution of the data and the prediction result, which can be explicitly represented as $P(Y, G|E) = P(G|E)P(Y|G, E)$. If we employ $D_{tr}(E)$ to represent the training data distribution for unobserved environments, the GNN-based model, when faced with OOD data, can rewrite Eq. (1) as:

$$\arg\min_\theta \ \mathbb{E}_{e\sim D_{tr}(E),(\mathcal{G}_u,y_u)\sim P(G,Y|E=e)}[l(f_\theta(\mathcal{G}_u;\theta),y_u)|e], \quad (6)$$

Eq. (6) shows that environment $E$ affects the data generation used for training the GNN-based recommendation model.

3.1.2 **Confounding Effect of E**. Figure 2 (a) and Figure 2 (b) illustrate the causal relationships in data generation and model training for graph-based recommendation algorithms. $E$ acts as the confounder and directly optimizing $P(Y|G)$ leads the GNN-based recommendation model to learn the shortcut predictive relationship between $\mathcal{G}_u$ and $y_u$, which is highly correlated with the environment $E$. During the model training process, there is a tendency to use this easily captured shortcut relationship to model user preferences. However, this shortcut relationship is highly sensitive to the environment $E$. When the environment of the test set is different from that of the training set (i.e., $D_{tr}(E) \neq D_{ts}(E)$), this relationship becomes unstable and invalid. The recommendation model that excessively learns environment-sensitive relationships in the training data will struggle to accurately model user preferences when faced with OOD data during the testing phase, resulting in decreased recommendation accuracy.

3.1.3 **Intervention**. Through the above analysis, we can improve the generalization ability of GNN-based recommendation models by guiding the model to uncover stable predictive relationships behind the training data, specifically those that are less sensitive to environmental changes. Thus, we can eliminate the influence of environmental confounders on model predictions. Specifically, we learn stable correlations between user item interaction $\mathcal{G}_u$ and ground truth $y_u$ by optimizing $P_\theta(Y|do(G))$ instead of $P_\theta(Y|G)$. In causal theory, the $do$-operation signifies removing the dependencies between the target variable and other variables. As shown in Figure 2 (c), by cutting off the causal relationship between the environment variables and the user interaction graph, the model no longer learns the unstable correlations between $\mathcal{G}_u$ and $y_u$. The $do$-operation simulates the generation process of the interaction graph $G$, where environmental factors do not influence the user-item interactions. This operation blocks the unstable backdoor path $G \leftarrow E \rightarrow Y$, enabling the GNN-based recommendation model to capture the desired causal relationship that remains invariant under environmental changes.

Theoretically, $P_\theta(Y|do(G))$ can be computed through randomized controlled trials, which involve randomly collecting new data from any possible environment to eliminate environmental bias. However, such physical interventions are challenging. For instance, in a short video recommendation setting, it is impossible to expose all short videos to a single user, and it is also impractical to control the environment of data interactions. In this paper, we achieve a statistical estimation of $P_\theta(Y|do(G))$ by leveraging backdoor adjustment. We have:

$$P_\theta(Y|do(G)) = \mathbb{E}_{e\sim D_{tr}(E)}[P_\theta(Y|G, E, I)]. \quad (7)$$

The derivation process is shown in Appendix B.1. Through the aforementioned backdoor adjustment, the influence of the environment $E$ on the generation of $G$ can be eliminated, enabling the model to learn correlations independent of the environment. However, in recommendation scenarios, environmental variables are typically unobservable or undefined, and their prior distribution $P(E = e)$ cannot be computed. Therefore, directly optimizing the Eq. (7) is challenging.

### 3.2 Model Instantiations

3.2.1 **Environment Inference**. This work introduces a variational inference method and proposes a variational inference-based environment instantiation mechanism. The core idea is to use variational inference to approximate the true distribution of environments and generate environment pseudo-labels as latent variables. The following tractable evidence lower bound (ELBO) can be obtained as the learning objective:

$$\log P_\theta(Y|do(G)) \geq \mathcal{L}_{envInf} = \mathbb{E}_{Q_\phi(E|G,I)}[\log P_\theta(Y|G, E, I)] \\ -D_{KL}(Q_\phi(E|G, I) \parallel P_\theta(E)), \quad (8)$$

where $Q_\phi(E|G, I)$ denotes environment estimation, which draws samples from the true distribution of the environment $E$. $D_{KL}$ represents the Kullback–Leibler (KL) divergence of the volitional distribution $Q_\phi(E|G, I)$ and the prior distribution $P_\theta(\text{E})$. $P_\theta(Y|G, E, I)$ is the graph representation learning module that employs the user-item interaction graph and the node attributes of users and items as input to learn invariant representations. Section 3.2.2 will provide a detailed introduction to the graph representation learning module. The derivation process of Eq. (8) is displayed in Appendix B.2.

3.2.2 **Invariant Representation Learning**. This section mainly consists of an environment generator, a diffusion-based graph representation learning module, and a recommendation module. Next, we will detail how they collaborate to enhance the generalizability of GNN-based models on OOD data and improve recommendation accuracy.

**Environment Generator**. In real-world recommendation scenarios, training datasets are collected in various environments. However, for a single user-centric interaction graph, the training dataset comes from a single environment. We need to learn environment-invariant correlations from training data originating from different environments to achieve the generalization capability of GNN-based recommendation models under distribution shifts. To circumvent this dilemma, this paper designs an environment generator $g_{\omega_k}(\cdot)(1 \leq k \leq K)$, which takes the user's original interaction graph $G$ as input and generates a set of $K$ interaction graphs $\{G_i\}_{i=1}^K$ to simulate training data from different environments. The optimization objective is expressed as follows:

$$\mathcal{L}_{generator} = \left[\text{Var}(\mathcal{L}(g_{\omega_k}(G)) : 1 \leq k \leq K)\right], \quad (9)$$

where $Var(\cdot)$ denotes the variance and $\mathcal{L}(\cdot)$ is the loss function. Following existing work [41], we modify the graph structure by adding and removing edges. Given a Boolean matrix $B_k$, the adjacency matrix $A$ of the graph, and its complement $A'$, the $k$-th

**Figure 3: Overall framework illustration of the proposed CausalDiffRec model.**

generated view for the original view is $A_k = A + B_k \odot (A - A')$. Since $B_k$ is a discrete matrix and not differentiable, it cannot be optimized directly. To address this issue, we borrow the idea from [41] and use reinforcement learning to treat graph generation as a decision process and edge editing as actions. Specifically, for view $k$, we consider a parameter matrix $\theta_k = \{\theta_{nm}^k\}$. For the $n$-th node, the probability of exiting the edge between it and the $m$-th node is given by:

$$h(a_{nm}^k) = \frac{\exp(\theta_{nm}^k)}{\sum_{m'=1}^{m'=m} \exp(\theta_{nm'}^k)}. \tag{10}$$

We then sample $s$ actions $\{b_{nmt}^k\}_{t=1}^{s}$ from a multinomial distribution $M(h(\alpha_{n1}^k), \ldots, h(\alpha_{nm}^k))$, which give the nonzero entries in the $n$-th row of $B_k$. The reward function $R(G_k)$ can be defined as the inverse loss. We can use the reinforcement algorithm to optimize the generator with the gradient:

$$\nabla_{\theta_k} \log h_{\theta_k}(A_k) R(G_k), \tag{11}$$

where $\theta_k$ is the model parameters and $h_{\theta_k}(A_k) = \prod_n \prod_{t=1}^{s} h(b_{nmt}^k)$. Optimizing Eq. (9) ensures that the generated graphs have large differences.

**Causal Diffusion**. Given the generated interaction graph $G_k = (A_k, I)$ where $A$ is the adjacency matrix, and $I$ is the feature matrix of users or items, instead of directly using $G_k$ as input for diffusion, we use the encoder from the Variational Graph Autoencoder (VGAE) to compress $G_k$ to a low-dimensional vector $\mathbf{x}_k^0 \sim \mathcal{N}(\mu_k, \sigma_k)$ for subsequent environment inference and graph invariant representation learning. The encoding process is as follows:

$$q_\psi(\mathbf{x}_k^0 | A_k, I) = N(\mathbf{x}_k^0 | \mu_k, \sigma_k), \tag{12}$$

where $\mu_k = GCN_\mu(A_k, I)$ is matrix of mean vectors and $\sigma_k = GCN_\sigma(A_k, I)$ denotes standard deviation. The $GCN(\cdot)$ is the graph convolution network in the graph variational autoencoder. According to the reparameterization trick, $\mathbf{x}_k^0$ can be calculated as follows:

$$\mathbf{x}_k^0 = \mu_k + \sigma_k \odot \epsilon, \tag{13}$$

where $\epsilon \sim N(0, I)$ and $\odot$ is the element product. Latent embedding $\mathbf{x}_k^0$ will be used as the input for the environment inference module to generate environment pseudo-labels. Meanwhile, $\mathbf{x}_k^0$ will do the forward and reverse processes in the latent space to learn the user/item embeddings in DDPM. The forward process can be

calculated as:

$$q(\mathbf{x}_k^{1:T} | \mathbf{x}_k^0) = \prod_{t=1}^{T} q(\mathbf{x}_k^t | \mathbf{x}_k^{t-1}). \tag{14}$$

After obtaining the environment approximation variable $z_{causal} = Q(E|G_k, I)$ according to Eq. (8), the pair of latent variables $(z_{causal}, \mathbf{x}_k^T)$ to learn the invariant graph representation. We approximate the inference distribution by parameterizing the probabilistic decoder through a conditional DDPM $p_\theta(\mathbf{x}_k^{t-1}|\mathbf{x}_k^T, z_{causal})$. Using DDPM, the forward process is entirely deterministic except for $t = 1$. We define the joint distribution of the reverse generative process as follows:

$$p_\theta(\mathbf{x}_k^{0:T} | z_{causal}) = p(\mathbf{x}_k^T) \prod_{t=1}^{T} p_\theta(\mathbf{x}_k^{t-1} | \mathbf{x}_k^t, z_{causal}). \tag{15}$$

$$p_\theta(\mathbf{x}_k^{t-1}|\mathbf{x}) = \mathcal{N}\left(\mathbf{x}_k^{t-1}; \mu_\theta(\mathbf{x}_k^t, z_{causal}, t), \Sigma_\theta(\mathbf{x}_k^t, z_{causal}, t)\right). \tag{16}$$

The loss function in Eq. (5) can be rewritten as:

$$\mathcal{L}_{\text{Invsample}} = \mathbb{E}_{t, \mathbf{x}_0, \epsilon_t}\left[\left\|\epsilon_t - \epsilon_\theta\left(\mathbf{x}_k^t, z_{causal}, t\right)\right\|^2\right], \tag{17}$$

where $\mathbf{x}_k^t = \sqrt{\alpha_t}\mathbf{x}_k^0 + \sqrt{1 - \alpha_t}\epsilon$. After obtaining the reconstructed output vector $\mathcal{R} = \mathbf{x}_k^{0'}$ from DDPM, it will be used as the input for the decoder of the variational graph decoder, which then reconstructs the input graph. The entire process is illustrated as follows:

$$\hat{A}_k = \phi(\mathcal{R}\mathcal{R}^\top), \tag{18}$$

where $\phi(\cdot)$ is the activation function (the sigmoid function is used in this paper). The VGAE is optimized by the variational lower bound:

$$\mathcal{L}_{VGAE} = \mathbb{E}_{q_\psi(\mathbf{x}_k^0|A_k, I)}\left[\log p_\theta(\hat{A}_k|\mathbf{x}_k^0)\right] \\ - D_{KL}\left(q_\psi(\mathbf{x}_k^0|A_k, I) \| p(\mathbf{x}_k^0)\right). \tag{19}$$

**Prediction and Joint Optimization**. Using the well-trained diffusion model to sample the final embeddings for user preference modeling:

$$\hat{r}_{u,i} = e_u^\top e_i, \tag{20}$$

where $e_u$ and $e_i$ denote the final user embedding w.r.t $u$-th user and item embedding w.r.t $i$-th item, respectively. Without loss of generality, LightGCN is used as the recommendation backbone, and

Bayesian Personalized Ranking (BRP) loss is employed to optimize the model parameters:

$$\mathcal{L}_{rec} = \sum_{u,v^+,v^-} -\log \sigma(\hat{r}_{u,v^+} - \hat{r}_{u,v^-}), \tag{21}$$

where $(u, v^+, v^-)$ is a triplet sample for pairwise recommendation training. $v^+$ represents positive samples from which the user has interacted, and $v^-$ are the negative samples that are randomly drawn from the set of items with which the user has not interacted, respectively. We use a joint learning strategy to optimize CausalDiffRec:

$$\begin{aligned}\mathcal{L} =&\mathcal{L}_{rec} + \lambda_1 \cdot \mathcal{L}_{generator} \\ &+ \lambda_2 \cdot (\mathcal{L}_{VGAE} + \mathcal{L}_{Invsample}) + \lambda_3 \cdot \mathcal{L}_{envInf},\end{aligned} \tag{22}$$

where $\lambda_1$, $\lambda_2$, and $\lambda_3$ are hyper-parameters. We provide rigorous theoretical proof in appendix A that optimizing the loss function in Eq. (22) can encourage the model to learn environment-invariant graph representations, thereby achieving generalization in out-of-distribution data recommendations. Model complexity analysis is provided in Appendix F.

## 4 Experiments

In this section, we conducted extensive experiments to validate the performance of CausalDiffRec and address the following key research questions:

- **RQ1**: How does CausalDiffRec compare to the state-of-the-art strategies in both OOD and IID test evaluations?
- **RQ2**: Are the proposed components of CausalDiffRec effective for OOD generalization?
- **RQ3**: How do hyperparameter settings affect the performance of CausalDiffRec?

### 4.1 Experimental Settings

**Datasets.** We evaluate the performance of our proposed CausalDiffRec method under three common data distribution shifts across four real-world datasets: Food[1], KuaiRec[2] Yelp2018[3], and Douban[4] comprise raw data from the Douban system. Detailed statistics of the datasets are presented in Table 1. The detailed information and processing specifics of the dataset can be found in Appendix C.

**Baselines**. We compare the CusalDiffRec with the state-of-the-art models: LightGCN[40], SGL [40], SimGCL[49], LightGCL[1], InvPref [39], InvCF [54], AdvDrop [52], AdvInfo [53], and DR-GNN [28]. Appendix E presents the detailed information of the baselines.

### 4.2 Overall Performance (RQ1)

This section compares CausalDiffRec's performance and baselines under various data shifts and conducts a performance analysis.

**Evaluation on temporal shift**: Table 1 shows that CausalDiffRec significantly outperforms SOTA models on the Food dataset, with improvements of 1.99%, 24.89%, 6.03%, and 9.68% in Recall and NDCG. This indicates CausalDiffRec's effectiveness in handling temporal shift. DRO also excels in this area, with a 15% improvement over LightGCN in NDCG@20, due to its robust optimization across various data distributions. CDR surpasses GNN-based models

---

[1]https://www.aclweb.org/anthology/D19-1613/
[2]https://kuairec.com
[3]https://www.yelp.com/dataset
[4]https://www.kaggle.com/datasets/

thanks to its temporal VAE-based architecture, capturing preference shifts from temporal changes.

**Evaluation on exposure shift**: In real-world scenarios, only a small subset of items is exposed to users, leading to non-random missing interaction records. Using the fully exposed KuaiRec dataset, CausalDiffRec consistently outperforms baselines, with improvements ranging from 6.90% to 28.83%, indicating its capability to handle exposure bias. DRO and AdvInfoNce also show superior performance in NDCG and Recall metrics, enhancing the generalization of GNN-based models and demonstrating robustness compared to LightGCN.

**Evaluation on popularity shift**. We compare model performance on the Yelp2018 and Douban datasets, showing that our model significantly outperforms the baselines. On Douban, CausalDiffRec achieves 8.22% to 17.96% improvement, and on Yelp2018, the improvements range from 11.24% to 36.73%. Methods using contrastive learning (e.g., SimGCL, LightGCL, AdvInfoNce) outperform other baselines in handling popularity shifts. This is because the InfoNCE loss helps the model learn a more uniform representation distribution, reducing bias towards popular items. InvPref performs best among the baselines on Yelp2018, using clustering for contextual labels, unlike our variational inference approach. Our method, tailored for graph data, aggregates neighbor information for better recommendation performance than matrix factorization-based methods.

Additionally, in Table 4, we report the performance of CausalDiffRec compared to several baseline models that use LightGCN as the backbone. From the table, we observe the following: 1) These baseline models outperform LightGCN on IID datasets; 2) CausalDiffRec outperforms all baseline models across all metrics. This indicates that CausalDiffRec also performs well on IID datasets. We attribute the performance improvement to our use of data augmentation and the incorporation of auxiliary information in modeling user preferences.

In summary, the analysis of experimental results demonstrates that our proposed CausalDiffRec can handle different types of distribution shifts and achieve good generalization.

### 4.3 In-depth Analysis (RQ2)

In this section, we conduct ablation experiments to study the impact of each component of CausalDiffRec on recommendation performance. The main components include the environment generator module and the environment inference module. Additionally, we use t-SNE to visualize the item representations captured by the baseline model and CausalDiffRec, to compare the models' generalization capabilities on OOD data.

**Ablation studies**. Table 2 presents the results of the ablation study that compares LightGCN, CausalDiffRec, and its two variants: 'w/o Gen.' (without the environment generator) and 'w/o Env.' (without the environment inference). The results show that removing these modules causes a significant drop in all metrics across four datasets. For example, on Yelp2018, Recall, and NDCG decreased by 148.15% and 126.83%, respectively, demonstrating the effectiveness of CausalDiffRec based on invariant learning theory for enhancing recommendation performance on OOD datasets. Additionally, even without the modules, CausalDiffRec still outperforms LightGCN on

**Table 1: The performance comparison between the baselines and CausalDiffRec on the four datasets with three data distribution shifts. The best results are highlighted in bold, and the second-best results are underlined. 'Impro.' denotes the relative improvements of CausalDiffRec over the second-best results.**

| Dataset | Metric | LightGCN | SGL | SimGCL | LightGCL | InvPref | InvCF | CDR | AdvDrop | AdvInfo | DR-GNN | Ours | Impro. |
|---------|--------|----------|-----|--------|----------|---------|-------|-----|---------|---------|--------|------|--------|
| Food | R@10 | 0.0234 | 0.0198 | 0.0233 | 0.0108 | 0.0029 | 0.0382 | 0.0260 | 0.0240 | 0.0227 | 0.0266 | **0.0281** | 5.63% |
| | N@10 | 0.0182 | 0.0159 | 0.0186 | 0.0101 | 0.0014 | 0.0237 | 0.0195 | 0.0251 | 0.0135 | 0.0205 | **0.0296** | 17.93% |
| | R@20 | 0.0404 | 0.0324 | 0.0414 | 0.0181 | 0.0294 | 0.0392 | 0.0412 | 0.0371 | 0.0268 | 0.0436 | **0.0464** | 6.42% |
| | N@20 | 0.0242 | 0.0201 | 0.0249 | 0.0121 | 0.0115 | 0.0240 | 0.0254 | 0.0237 | 0.0159 | 0.0279 | **0.0306** | 9.68% |
| KuaiRec | R@10 | 0.0742 | 0.0700 | 0.0763 | 0.0630 | 0.0231 | 0.1023 | 0.0570 | 0.1014 | 0.1044 | 0.0808 | **0.1116** | 6.90% |
| | N@10 | 0.5096 | 0.4923 | 0.5180 | 0.4334 | 0.2151 | 0.2242 | 0.2630 | 0.3290 | 0.4302 | 0.5326 | **0.6474** | 21.55% |
| | R@20 | 0.1120 | 0.1100 | 0.1196 | 0.1134 | 0.0478 | 0.1034 | 0.0860 | 0.1214 | 0.1254 | 0.1266 | **0.1631** | 28.83% |
| | N@20 | 0.4268 | 0.4181 | 0.4446 | 0.4090 | 0.2056 | 0.2193 | 0.2240 | 0.3289 | 0.4305 | 0.4556 | **0.5392** | 18.35% |
| Yelp2018 | R@10 | 0.0014 | 0.0027 | 0.0049 | 0.0022 | 0.0049 | 0.0004 | 0.0011 | 0.0027 | 0.0047 | 0.0044 | **0.0067** | 36.73% |
| | N@10 | 0.0008 | 0.0017 | 0.0028 | 0.0015 | 0.0030 | 0.0026 | 0.0006 | 0.0017 | 0.0024 | 0.0029 | **0.0039** | 30.00% |
| | R@20 | 0.0035 | 0.0051 | 0.0106 | 0.0054 | 0.0108 | 0.0013 | 0.0016 | 0.0049 | 0.0083 | 0.0076 | **0.0120** | 11.65% |
| | N@20 | 0.0016 | 0.0026 | 0.0047 | 0.0026 | 0.0049 | 0.0008 | 0.0008 | 0.0024 | 0.0038 | 0.0041 | **0.0055** | 11.11% |
| Douban | R@10 | 0.0028 | 0.0022 | 0.0086 | 0.0070 | 0.0052 | 0.0030 | 0.0014 | 0.0051 | 0.0076 | 0.0028 | **0.0094** | 9.30% |
| | N@10 | 0.0015 | 0.0013 | 0.0045 | 0.0038 | 0.0026 | 0.0012 | 0.0007 | 0.0021 | 0.0042 | 0.0011 | **0.0050** | 11.11% |
| | R@20 | 0.0049 | 0.0047 | 0.0167 | 0.0113 | 0.0093 | 0.0033 | 0.0200 | 0.0046 | 0.0103 | 0.0038 | **0.0197** | 17.96% |
| | N@20 | 0.0019 | 0.0020 | 0.0073 | 0.0050 | 0.0038 | 0.0013 | 0.0019 | 0.0021 | 0.0053 | 0.0015 | **0.0079** | 8.22% |

**Table 2: Outcomes from ablation studies on four datasets. The top-performing results are highlighted in bold, while those that are second-best are underlined.**

| Dataset | Ablation | R@10 | R@20 | N@10 | N@20 |
|---------|----------|------|------|------|------|
| Food | LightGCN | 0.0234 | 0.0404 | 0.0182 | 0.0242 |
| | w/o Gen. | 0.0165 | 0.0259 | 0.0114 | 0.0148 |
| | w/o Env. | 0.0084 | 0.0144 | 0.0077 | 0.0098 |
| | CausalDiffRec | **0.0251** | **0.0409** | **0.0296** | **0.0306** |
| KuaiRec | LightGCN | 0.0808 | 0.1266 | 0.5326 | 0.4556 |
| | w/o Gen. | 0.0966 | 0.1571 | 0.0445 | 0.3078 |
| | w/o Env. | 0.0047 | 0.1740 | 0.0697 | 0.0784 |
| | CausalDiffRec | **0.1116** | **0.1631** | **0.0674** | **0.5392** |
| Yelp2018 | LightGCN | 0.0014 | 0.0035 | 0.0008 | 0.0016 |
| | w/o Gen. | 0.0041 | 0.0054 | 0.0037 | 0.0042 |
| | w/o Env. | 0.0027 | 0.0058 | 0.0042 | 0.0043 |
| | CausalDiffRec | **0.0067** | **0.0120** | **0.0039** | **0.0055** |
| Douban | LightGCN | 0.0028 | 0.0049 | 0.0015 | 0.0019 |
| | w/o Gen. | 0.0044 | 0.0079 | 0.0030 | 0.0045 |
| | w/o Env. | 0.0044 | 0.0070 | 0.0023 | 0.0031 |
| | CausalDiffRec | **0.0094** | **0.0197** | **0.0050** | **0.0079** |

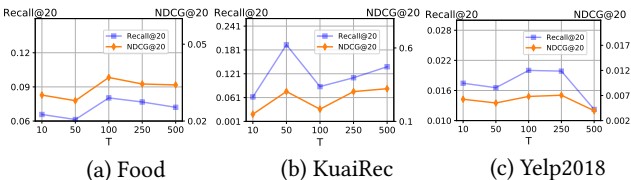

(a) Food        (b) KuaiRec        (c) Yelp2018

**Figure 4: Effects of the number of diffusion steps T.**

and CausalDiffRec on Douban and Yelp2018 datasets to better observe our model's ability to handle distribution shifts. Following previous work [28], we recorded the popularity of each item in the training set and designated the top 10% most popular items as 'popular items' and the bottom 10% as 'unpopular items'. It is obvious that the embeddings of popular and unpopular items learned by LightGCN still exhibit a gap in the representation space. In contrast, the embeddings learned by CausalDiffRec are more evenly distributed within the same space. This indicates that CausalDiffRec can mitigate the popularity shift caused by popular items. Additionally, we found that the embeddings of popular and unpopular items learned by SimGCL are more evenly distributed compared to LightGCN. This is because contrastive learning can learn a uniform representation distribution.

## 4.4 Hyperparameter Investigation (RQ3)

**Effect of Diffusion Step T**. We conducted experiments to investigate the impact of the number of diffusion steps on performance; in CausalDiffRec, we used the same number of steps in the forward and reverse phases. we compare the performance with $T$ changing from 10 to 500. We present the results in Figure 4, and we have the following findings:

- When the number of steps is chosen within the range {10, 50, 100}, CausalDiffRec achieves the best performance across all

popularity shift datasets (Yelp2018 and Douban) due to the effectiveness of data augmentation and environment inference. However, on the Food and KuaiRec datasets, removing either module results in worse performance than LightGCN, likely due to multiple biases in these datasets. Without one module, the model struggles to handle multiple data distributions, leading to a performance drop. Overall, the ablation experiments highlight the importance of all modules in CausalDiffRec for improving recommendation performance and generalizing on OOD data.

**Visualization analysis**. In Figure 6 and Figure 7, we used t-SNE to visualize the item representations learned by LightGCN, SimGCL,

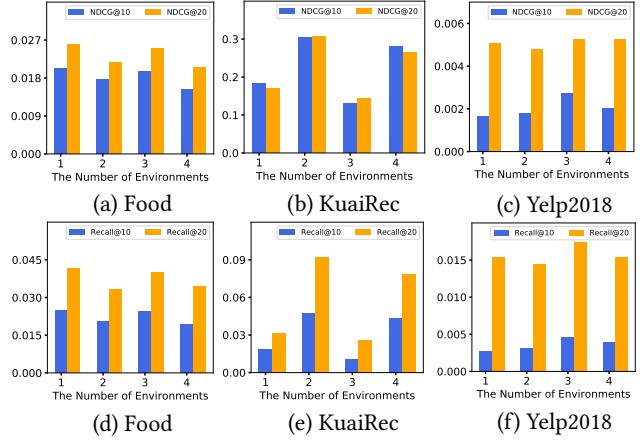

**Figure 5: Effects of the number of environments K.**

datasets. We find that appropriately increasing the number of steps significantly improves Recall@20 and NDCG@20. These performance enhancements are mainly attributed to the diffusion enriching the representation capabilities of users and items.

- Nevertheless, as we continue to increase the number of steps, the model will face overfitting issues. For example, on the food and yelp2018 datasets, Recall@20 and NDCG@20 consistently decrease. Although there is an upward trend on KuaiRec, the optimal solution is not achieved. Additionally, it is evident that more steps also lead to longer training times. We should carefully adjust the number of steps to find the optimal balance between enhancing representation ability and avoiding overfitting.

**Effect of the number of Environments**. Figure 5 shows the impact of the number of environments on the model's performance. We can see that as the number of environments increases, the performance of CausalDiffRec improves across the three datasets. This indicates that more environments help enhance the model's generalization on OOD (Out-of-Distribution) data. However, as the number of environments further increases, performance declines, which we believe is due to the model overfitting to too many environments.

## 5 Related Work

### 5.1 GNN-based Recommendation

Recent developments in graph-based recommender systems have leveraged graph neural networks to model user-item interactions as a bipartite graph, enhancing recommendation accuracy through complex interaction capture [9, 35, 42]. Notably, LightGCN focuses on neighborhood aggregation without additional transformations, while other approaches employ attention mechanisms to prioritize influential interactions [3, 6, 25, 29, 36, 37]. Further research explores non-Euclidean spaces like hyperbolic space to better represent user-item relationships [27, 55]. Knowledge graphs also enhance these systems by integrating rich semantic and relational data directly into the recommendation process [2, 34]. Despite these advancements, graph-based systems often struggle with out-of-distribution data due to the IID assumption and are challenged

by multiple distribution shifts [15, 19, 40, 45, 49]. Additionally, contrastive learning methods in these systems rely on a fixed paradigm that lacks robust theoretical support, limiting adaptability to varied data shifts. However, the aforementioned models are trained on datasets where the training and test data distributions are drawn from the same distribution, leading to generalization failure when facing OOD data.

### 5.2 Diffusion based Recoomendation

The integration of diffusion processes into recommender systems leverages diffusion mechanisms to model dynamic propagation of user preferences and item information through interaction networks, enhancing recommendation accuracy and timeliness [16, 21, 22, 24, 26, 33, 44]. These models capture evolving user behaviors and have shown potential in various recommendation contexts, from sequential recommendations to location-based services. For instance, DiffRec [33] applies diffusion directly for recommendations, while Diff-POI [26] models location preferences. Furthermore, approaches like DiffKG [16] and RecDiff [22] utilize diffusion for denoising entity representations in knowledge graphs and user data in social recommendations, respectively, enhancing the robustness and reliability of the systems. These studies underscore diffusion's suitability for advanced representation learning in recommender systems. However, these methods cannot solve the OOD problem.

### 5.3 Out-of-Distribution Recommendation

Researchers have focused on recommendation algorithms for out-of-distribution (OOD) data. COR [31] infers latent environmental factors in OOD data. CausPref [11] learns invariant user preferences and causal structures using anti-preference negative sampling. CaseQ [46] employs backdoor adjustment and variational inference for sequential recommendations. InvPref [39] separates invariant and variant preferences by identifying heterogeneous environments. However, these methods don't directly apply to graph-based recommendation models and fail to address OOD in graph structures. AdaDrop [52] uses adversarial learning and graph neural networks to enhance performance by decoupling user preferences. DRO [28] integrates Distributionally Robust Optimization into Graph Neural Networks to handle distribution shifts in graph-based recommender systems. Distinct from these GNN-based methods, this paper explores how to use the theory of invariant learning to design GNN-based methods with good generalization capabilities.

## 6 Conclusion

This paper introduces CausalDiffRec, an innovative GNN-based model designed for OOD recommendation. CausalDiffRec aims to learn environment-invariant graph representations to improve model generalization on OOD data. It utilizes the backdoor criterion from causal inference and variational inference to mitigate environmental confounders, alongside a diffusion-based sampling strategy. Rooted in invariant learning theory, we theoretically demonstrate that optimizing CausalDiffRec's objective function enhances its ability to identify invariant graph representations, boosting generalization on OOD data. Experiments on four real-world datasets show CausalDiffRec surpasses baseline models, with ablation studies confirming its effectiveness.

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

## A   Theoretical Proof

CausalDiffRec aims to learn the optimal generator $F^*(\cdot)$ as stated in the assumption in section 2.3, thereby obtaining invariant graph representations to achieve OOD generalization in recommendation performance under data distribution shifts. Before starting the theoretical derivation, let's do some preliminary work. For the convenience of theoretical proof, we rewrite Eq. (22) as:

$$\arg\min_\theta \ (\mathcal{L}_{\text{task}} + \mathcal{L}_{\text{infer}}), \tag{23}$$

where $\mathcal{L}_{task} = \mathcal{L}_{rec} + \mathcal{L}_{generator} + \mathcal{L}_{VGAE} + \mathcal{L}_{Invsample}$ and $\mathcal{L}_{infer} = \mathcal{L}_{envInf}$, and for the derivation convenience, we temporarily ignore the penalty coefficient. The $\mathcal{L}_{\text{task}}$ and $\mathcal{L}_{\text{infer}}$ can be further abstracted as:

$$
\begin{aligned}
\mathcal{L}_{\text{task}} &= \arg\min_\theta \mathbb{E}_{e \sim D_{tr}(E), (\mathcal{G}_u, y_u) \sim P(Y, G|E=e)} \\
&\quad [l(f_\theta(\mathcal{G}_u; \theta), y_u)] \\
\mathcal{L}_{\text{infer}} &= \min_{q(Y|z_{causal})} \text{Var}\{\mathbb{E}_{e \sim D_{tr}(E), (\mathcal{G}_u, y_u) \sim P(Y, G|E=e)} \\
&\quad [l(f_\theta(\mathcal{G}_u; \theta), y_u)|do(\mathcal{G}_u)]\}.
\end{aligned}
\tag{24}
$$

We follow the proof technique from [51] and show the optimality of the Eq. (23) with the following two propositions that can achieve OOD recommendation.

PROPOSITION A.1. *Minimizing Eq. (23) promotes the model's adherence to the Invariance Property and the Sufficient Condition outlined in* **Assumption** *(in Sec. 2.3).*

PROPOSITION A.2. *Optimizing Eq. (23) corresponds to minimizing the upper bound of the OOD generalization error described in Eq. (6).*

Proposition A.1 and Proposition A.2, respectively, avoid strong hypotheses and ensure that the OOD generalization error bound of the learned model is within the expected range. In fact, this can also be explained from the perspective of the SCM model in Figure 2. Optimizing Eq. (6) eliminates the negative impact of unstable correlations learned by the model, which are caused by latent environments, on modeling user preferences. At the same time, it enhances the model's ability to learn invariant causal features across different latent environments. Proofs for Proposition A.1 and Proposition A.2 are shown as follows:

Before starting the proof, we directly follow previous work [4, 18, 41, 48, 51] to propose the following lemma, using information theory to interpret the invariance property and sufficient condition in **Assumption** and to assist in the proof of Proposition A.1. Using the Mutual Information $\mathbb{I}(;)$, the invariance property and sufficient condition in **Assumption** can be equivalently expressed as follow lemma:

LEMMA A.3. *(1) Invariance:* $\forall e \in D(E), P_\theta(Y|\mathcal{P}_{Inv}^*, E = e, I) = P(Y|\mathcal{P}_{Inv}^*, I) \Leftrightarrow \mathbb{I}(Y; E|\mathcal{P}_{Inv}^*, I) = 0$ *where* $\mathcal{P}_{Inv}^* = F^*(G)$. *(2) Sufficiency:* $\mathbb{I}(Y; \mathcal{P}_{Inv}^*, I)$ *is maxmized.*

For the invariance property, it is easy to get the following equation:

$$
\begin{aligned}
&\mathbb{I}(Y; E|\mathcal{P}_{\text{Inv}}^*, I) \\
&= \mathbb{E}_{\mathcal{P}_{\text{Inv}}^*, I} \left[ \mathbb{D}_{\text{KL}} \left( P(Y, E|\mathcal{P}_{\text{Inv}}^*, I) \| P(Y|\mathcal{P}_{\text{Inv}}^*, I) P(E|\mathcal{P}_{\text{Inv}}^*, I) \right) \right]
\end{aligned}
\tag{25}
$$

For the sufficient condition, we employ the method of contradiction and prove it through the following two steps:

**First**, we prove that for $Y, \mathcal{P}_{Inv}^*$, and $I$ satisfying $\mathcal{P}_{Inv}^* = \arg\max_{\mathcal{P}_{Inv}} \mathbb{I}(Y; \mathcal{P}_{Inv}, I)$, they also satisfy that $\mathbb{I}(Y; \mathcal{P}_{Inv}^*, I)$ is maximized. We leverage the method of contradiction to prove this. Assume $\mathcal{P}_{Inv}^* \neq \arg\max_{\mathcal{P}_{Inv}} \mathbb{I}(Y; \mathcal{P}_{Inv}, I)$, and there exists $\mathcal{P}_{Inv}' = \arg\max_{\mathcal{P}_{Inv}} \mathbb{I}(Y; \mathcal{P}_{Inv}, I)$, where $\mathcal{P}_{Inv}' \neq \mathcal{P}_{Inv}^*$. We can always find a mapping function $M$ such that $\mathcal{P}_{Inv}' = M(\mathcal{P}_{Inv}^*, R)$, where $R$ is a random variable. Then we have:

$$
\begin{aligned}
\mathbb{I}(Y; \mathcal{P}_{Inv}', I) &= \mathbb{I}(Y; \mathcal{P}_{Inv}^*, R, I) \\
&= \mathbb{I}(Y; \mathcal{P}_{Inv}^*, I) + \mathbb{I}(Y; R|\mathcal{P}_{Inv}^*, I).
\end{aligned}
\tag{26}
$$

Since $R$ is a random variable and does not contain any information about $Y$, we have $\mathbb{I}(Y; R|\mathcal{P}_{Inv}^*, I) = 0$. Therefore:

$$\mathbb{I}(Y; \mathcal{P}_{Inv}', I) = \mathbb{I}(Y; \mathcal{P}_{Inv}^*, I). \tag{27}$$

This leads to a contradiction.

**Next**, we prove that for $Y, \mathcal{P}_{Inv}^*$, and $I$ satisfying $\mathcal{P}_{Inv}^* = \arg\max_{\mathcal{P}_{Inv}} \mathbb{I}(Y; \mathcal{P}_{Inv}, I)$, they also satisfy that $\mathbb{I}(Y; \mathcal{P}_{Inv}^*, I)$ is maximized. Assume $\mathcal{P}_{Inv}^* \neq \arg\max_{\mathcal{P}_{Inv}} \mathbb{I}(Y; \mathcal{P}_{Inv}, I)$, and there exists $\mathcal{P}_{Inv}' = \arg\max_{\mathcal{P}_{Inv}} \mathbb{I}(Y; \mathcal{P}_{Inv}, I)$, where $\mathcal{P}_{Inv}' \neq \mathcal{P}_{Inv}^*$. We have the following inequality:

$$\mathbb{I}(Y; \mathcal{P}_{Inv}^*, I) \leq \mathbb{I}(Y; \mathcal{P}_{Inv}', I). \tag{28}$$

From this, we can deduce that:

$$\mathcal{P}_{Inv}' = \arg\max_{\mathcal{P}_{Inv}} \mathbb{I}(Y; \mathcal{P}_{Inv}, I), \tag{29}$$

where contradicts $\mathcal{P}_{Inv}^* = \arg\max_{\mathcal{P}_{Inv}} \mathbb{I}(Y; \mathcal{P}_{Inv}, I)$. Since the assumption leads to a contradiction, and the assumption does not hold. Therefore, $\mathcal{P}_{Inv}^* = \arg\max_{\mathcal{P}_{Inv}} \mathbb{I}(Y; \mathcal{P}_{Inv}, I)$ holds. This proves that $\mathbb{I}(Y; \mathcal{P}_{Inv}^*, I)$ is maximized. The lemma A.3 is complicated proven.

**Proof of Proposition** A.1. **First**, optimizing the first term $\mathcal{L}_{\text{task}}$ in Eq. (23) enables the model to satisfy the sufficient condition. Analyzing the SCM in Figure 2(c), we have the fact that $\max_{q(z|G,I)} \mathbb{I}(Y, z_{causal})$ is equivalent to $\min_{q(z|G,I)} \mathbb{I}(Y, G|Z_{causal})$, as we use $do(G)$ to eliminate the unstable correlations between Y and G caused by the latent environment. We have:

$$
\begin{aligned}
\mathbb{I}(Y, G|z_{causal}) &= D_{KL}(p(Y|G, E) \| p(Y|z_{causal}, E)) \\
&= D_{KL}(p(Y|G, E) \| p(Y|z_{causal})) \\
&\quad - D_{KL}(p(Y|z_{causal}, E) \| q(Y|z_{causal})) \\
&\leq D_{KL}(p(Y|G, E) \| p(Y|z_{causal})).
\end{aligned}
\tag{30}
$$

Based on the above derivation, we have:

$$
\begin{aligned}
&\mathbb{I}(Y, G|z_{causal}) \leq \\
&\min_{q(Y|z_{causal})} D_{KL}(p(Y|G, E) \| p(Y|z_{causal})).
\end{aligned}
\tag{31}
$$

Besides, we have:

$$D_{KL}(p(Y|G,E)\|p(Y|z_{causal}))$$

$$= \mathbb{E}_{e \in D_{tr}(E)}\mathbb{E}_{(G,Y)\sim p(G,Y|e)}$$

$$\mathbb{E}_{z_{causal}\sim q(z_{causal}|G,I)}\left[\log\frac{q(Y|G,e)}{p(Y|z_{causal})}\right]$$

$$\leq \mathbb{E}_{e \in D_{tr}(E)}\mathbb{E}_{(G,Y)\sim p(G,Y|e)}$$

$$\left[\log\frac{p(Y|G,e)}{\mathbb{E}_{z_{causal}\sim q(z_{causal}|G,I)}q(Y|z_{causal})}\right] \text{(Jensen Inequality)}. \tag{32}$$

Finally, we reach:

$$\min_{q(Y|z_{causal})} D_{KL}(p(Y|G,E)\|p(Y|z_{causal}))$$

$$\Leftrightarrow \arg\min_{\theta}\mathbb{E}_{e\sim D_{tr}(E),(\mathcal{G}_u,y_u)\sim P(Y,G|E=e)}[l(f_{\theta}(\mathcal{G}_u;\theta),y_u)]. \tag{33}$$

Thus, we have demonstrated that minimizing the expectation term ($\mathcal{L}_{task}$) in Eq. (23) is equivalent to minimizing the upper bound of $\mathbb{I}(Y;G \mid z_{causal})$. This results in maximizing $\mathbb{I}(Y;\mathcal{P}_{Inv}^*,I)$, thereby helping to ensure that the model satisfies the Sufficient Condition.

**Next**, we prove that optimizing the first term $\mathcal{L}_{task}$ in Eq. (23) enables the model to satisfy the Invariance Property. Similar to Eq. (30), we have:

$$\mathbb{I}(Y;E=e \mid z_{causal})$$

$$= D_{KL}(p(Y \mid z_{causal},e) \| p(Y \mid z_{causal}))$$

$$= D_{KL}(p(Y \mid z_{causal},E) \| \mathbb{E}_{e\in D(E)}[p(Y \mid z_{causal},e)])$$

$$= D_{KL}(q(Y \mid z_{causal}) \| \mathbb{E}_{e\in D(E)}[q(Y \mid z_{causal})])$$

$$- D_{KL}(q(Y \mid z_{causal}) \| p(Y \mid z_{causal},e))$$

$$- D_{KL}(\mathbb{E}_{e\in D(E)}[p(Y \mid z_{causal},e)] \| \mathbb{E}_{e\in D(E)}[q(Y \mid z_{causal})])$$

$$\leq D_{KL}(q(Y \mid z_{causal}) \| \mathbb{E}_{e\in D(E)}[q(Y \mid z_{causal})]). \tag{34}$$

Besides, the last term in Eq. (34) can be further expressed as:

$$D_{KL}(q(Y \mid z_{causal}) \| \mathbb{E}_{e\in D(E)}[q(Y \mid z_{causal})])$$

$$= \mathbb{E}_{e\in D_{tr}(E)}\mathbb{E}_{(G,Y)\sim p(G,Y|e)}\mathbb{E}_{z_{causal}\sim q(z_{causal}|G,I)}$$

$$\left[\log\frac{p(Y|G,e)}{\mathbb{E}_{e\in D(E)}q(Y|z_{causal})}\right] \text{(Jensen Inequality)} \tag{35}$$

$$\leq \mathbb{E}_{e\in D(E)}[|l(f_{\theta}(\mathcal{G}_u;\theta),y_u) - \mathbb{E}_{e\in D(E)}[l(f_{\theta}(\mathcal{G}_u;\theta),y_u)]|]$$

where the last term in Eq. (35) is the upper bound for the $D_{KL}(q(Y \mid z_{causal}) \| \mathbb{E}_{e\in D(E)}[q(Y \mid z_{causal})])$. Finally, we have:

$$\min_{q(Y|z_{causal})}D_{KL}(q(Y \mid z_{causal}) \| \mathbb{E}_{e\in D(E)}[q(Y \mid z_{causal})])$$

$$\Leftrightarrow \min_{q(Y|z_{causal})}\text{Var}\{\mathbb{E}_{e\sim D_{tr}(E),(\mathcal{G}_u,y_u)\sim P(Y,G|E=e)}$$

$$[l(f_{\theta}(\mathcal{G}_u;\theta),y_u)|do(\mathcal{G}_u)]\}. \tag{36}$$

Hence, minimizing the variance term ($\mathcal{L}_{risk}$) in Eq. (23) effectively reduces the upper bound of $\mathbb{I}(Y;E=e \mid z_{causal})$. Thereby ensuring the model adheres to the Invariance Property.

**Proof of Proposition** A.2. Optimizing Eq. (23) is tantamount to reducing the upper bound of the OOD generalization error in Eq. (6). Let $q(Y \mid G)$ represent the inferred variational distribution of the true distribution $p(Y \mid G,E)$. The OOD generalization error can

be quantified by the KL divergence between these two distributions:

$$D_{KL}(p(Y|G,E)\|q(Y|G))$$

$$= \mathbb{E}_{e\in D_{tr}(E)}\mathbb{E}_{(Y,G)\sim p(Y,G|E)}\log\frac{p(Y|G,E=e)}{q(Y|G)}. \tag{37}$$

Following previous work, we use information theory to assist in the proof of Proposition A.2. We propose the lemma A.4 to rewrite the OOD generalization, which is shown as follows:

LEMMA A.4. *The out-of-distribution generalization error is limited by:*

$$D_{KL}(p(Y|G,E)\|q(Y|G)) \leq D_{KL}[p(Y|G,E)\|q(Y|z_{causal})], \tag{38}$$

where $q(Y|z_{causal})$ is the inferred variational environment distribution. The proof of Lemma A.4 is shown as:

$$D_{KL}(p(Y|G,E=e)\|q(Y|z_{causal}))$$

$$= \mathbb{E}_{e\in D(E)}\mathbb{E}_{(Y,G)\sim p(G,Y|E=e)}\left[\log\frac{p(Y|G,E=e)}{q(Y|G)}\right]$$

$$= \mathbb{E}_{e\in D(E)}\mathbb{E}_{(Y,G)\sim p(G,Y|E=e)}$$

$$\left[\log\frac{p(Y|G,E=e)}{\mathbb{E}_{z_{causal}\sim q(z_{causal}|G,I)}q(Y|z_{causal})}\right] \tag{39}$$

$$\leq \mathbb{E}_{e\in D(E)}\mathbb{E}_{(Y,G)\sim p(G,Y|E=e)}$$

$$\mathbb{E}_{z_{causal}\sim q(z_{causal}|G,I)}\log\frac{p(Y|G,e)}{q(Y|z_{causal})}$$

$$= D_{KL}[p(Y|G,E)\|q(Y|z_{causal})],$$

The Lemma A.4 has been fully proven. Based on Lemma A.3 and Proposition A.1, the Eq. (23) can be adapted as:

$$\min_{q(z_{causal}|G,I),q(Y,z_{causal})}$$
$$D_{KL}(p(Y|G,E=e)\|q(Y|z_{causal})) + \mathbb{I}(Y,E=e|z_{causal}) \tag{40}$$

Hence, according to Lemma A.4, we confirm that minimizing Eq. (23) is equivalent to minimizing the upper bound of the OOD generalization error in Eq. (6), meaning that:

$$\text{argmin}_{\theta}(\mathcal{L}_{task} + \mathcal{L}_{infer}) \Leftrightarrow \min_{q(z_{causal}|G,I),q(Y,z_{causal})}$$

$$D_{KL}(p(Y|G,E=e)\|q(Y|z_{causal}))$$

$$+ \mathbb{I}(Y,E=e|z_{causal}) \text{ } (\mathbb{I}(Y,E=e|z_{causal}) \text{ is non} - negative)$$

$$\geq \min_{q(z_{causal}|G,I),q(Y,z_{causal})} \text{ } D_{KL}(p(Y|G,E=e)\|q(Y|z_{causal}))$$

$$\geq D_{KL}(p(Y|G,E)\|q(Y|G)). \tag{41}$$

The Proposition A.2 is completely proven.

# B  Derivation

## B.1  Derivation for Equation 7

$$P_{\theta}(Y|do(G))$$

$$= \sum_e P_{\theta}(Y|do(G),E=e,I)P_{\theta}(E=e|do(G))P_{\theta}(I)$$

$$= \sum_e P_{\theta}(Y|G,E=e,I)P_{\theta}(E=e|do(G))P_{\theta}(I)$$

$$= \sum_e P_{\theta}(Y|G,E=e,I)P_{\theta}(E=e)P_{\theta}(I) \tag{42}$$

$$= \sum_e P_{\theta}(Y|G,E=e,I)P_{\theta}(E=e,I)$$

$$= \mathbb{E}_{e\sim D_{tr}(E)}[P_{\theta}(Y|G,E,I)],$$

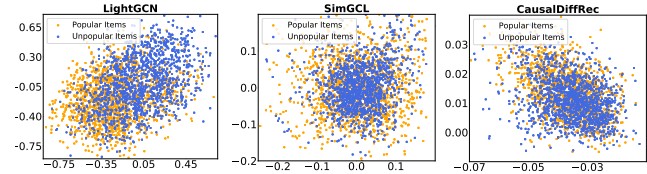

**Figure 6: Visualization of user embedding distributions using various methods on the Douban dataset. CausalDiffRec ensures that hot items and cold items have representations that are nearly co-located within the same space.**

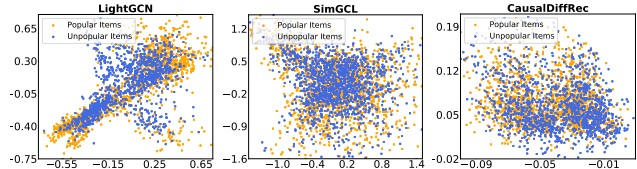

**Figure 7: Visualization of user embedding distributions using various methods on the Yelp2018 dataset. CausalDiffRec ensures that hot items and cold items have representations that are nearly co-located within the same space.**

**Table 3: Detailed statistics for each dataset.**

| Dataset | #Users | #Items | #Interactions | Density |
|---------|--------|--------|---------------|---------|
| Food | 7,809 | 6,309 | 216,407 | $4.4 \times 10^{-3}$ |
| KuaiRec | 7,175 | 10,611 | 1,153,797 | $1.5 \times 10^{-3}$ |
| Yelp2018 | 8,090 | 13,878 | 398,216 | $3.5 \times 10^{-3}$ |
| Douban | 8,735 | 13,143 | 354,933 | $3.1 \times 10^{-3}$ |

## B.2 Derivation for Equation 8

Taking the logarithm on both sides of Eq. (8) and according to Jensen's Inequality, we have:

$$
\begin{aligned}
&\log P_\theta(Y|do(G)) \\
&= \log \mathbb{E}_{e \sim D_{tr}(E)}[P_\theta(Y|G,E,I)] \\
&= \log \sum_e P_\theta(Y|G,E=e,I)P_\theta(E=e,I)\frac{Q_\phi(E=e|G,I)}{Q_\phi(E=e|G,I)} \\
&\geq \sum_e Q_\phi(E=e|G,I) \\
&\quad \log P_\theta(Y|G,E=e,I)P_\theta(E=e,I)\frac{1}{Q_\phi(E=e|G,I)} \quad (43) \\
&= \sum_e [Q_\phi(E=e|G,I)\log P_\theta(Y|G,E=e,I)- \\
&\quad \log \frac{Q_\phi(E=e|G,I)P_\theta(E=e,I)}{Q_\phi(E=e|G,I)}] \\
&= \mathbb{E}_{Q_\phi(E=e|G,I)}[\log P_\theta(Y|G,E=e,I)] \\
&\quad - D_{KL}(Q_\phi(E=e|G,I) \parallel P_\theta(E=e)).
\end{aligned}
$$

**Table 4: Model performance comparison on IID datasets.**

| Dataset | Ablation | R@10 | R@20 | N@10 | N@20 |
|---------|----------|------|------|------|------|
| KuaiRec | LightGCN | 0.0154 | 0.0174 | 0.0272 | 0.0210 |
| | AdvDrop | 0.0431 | 0.0258 | 0.0469 | 0.0276 |
| | AdvInfo | 0.0514 | 0.0298 | 0.0518 | 0.0300 |
| | DRO | 0.0307 | 0.0226 | 0.0505 | 0.0291 |
| | CausalDiffRec | **0.0567** | **0.0472** | **0.0634** | **0.0707** |
| Yelp2018 | LightGCN | 0.0023 | 0.0022 | 0.0039 | 0.0029 |
| | AdvDrop | 0.0061 | 0.0051 | 0.0063 | 0.0050 |
| | AdvInfo | 0.0052 | 0.0040 | 0.0062 | 0.0049 |
| | DRO | 0.0069 | 0.0550 | 0.0086 | 0.0065 |
| | CausalDiffRec | **0.0102** | **0.0120** | **0.0182** | **0.0134** |

## C Dataset detail

**Processing Details**. We retain only those users with at least 15 interactions on the Food dataset, at least 25 interactions on the Yelp2018 and Douban datasets, and items with at least 50 interactions on these datasets. For all three datasets, only interactions with ratings of 4 or higher are considered positive samples. For the KuaiRec dataset, interactions with a watch ratio of 2 or higher are considered positive samples.

We directly follow [28] to process the dataset above to construct three common types of out-of-distribution data:

- **Popularity shift**: We randomly select 20% of interactions to form the OOD test set, ensuring a uniform distribution of item popularity. The remaining data is split into training, validation, and IID test sets in a ratio of 7:1:2, respectively. This type of distribution shift is applied to the Yelp2018 and Douban datasets.
- **Temporal shift**: We sort the dataset by timestamp in descending order and designate the most recent 20% of each user's interactions as the OOD test set. The remaining data is split into training, validation, and IID test sets in a ratio of 7:1:2, respectively. The food dataset is used for this type of distribution shift.
- **Exposure shift**: In KuaiRec, the smaller matrix, which is fully exposed, serves as the OOD test set. The larger matrix collected from the online platform is split into training, validation, and IID test sets in a ratio of 7:1:2, respectively, creating a distribution shift.

## D Hyperparameter Settings

We implement our CausalDiffRec in Pytorch. All experiments are conducted on a single RTX-4090 with 24G memory. Following the default hyperparameter search settings of the baselines, we expand their hyperparameter search space and tune the hyperparameters. For our CausalDiffRec, we tune the learning rates in $\{1e-3, 1e-4, 1e-5\}$. The number of diffusion steps varies between 10 and 1000, and the diffusion embedding size is chosen in $\{8, 16, 32, 64\}$. Additional hyperparameter details are available in our released code.

## E Baselines

We compare the CusalDiffRec with the following state-of-the-art models:

**Algorithm 1** Training of CausalDiffRec under Multiple Environments

1: **Input:** The user-item interaction graph $G(\mathcal{V}, \mathcal{E})$ and node feature matrix $\mathcal{X}$; Using the $\omega$, $\theta_1$, and $\theta_2$ to initial environment generator $g_\omega(\cdot)$, environment $P_{\theta_1}(\cdot)$, and graph representation learner (ie., sampling approximator) $f_{\theta_2}(\cdot)$, respectively.
2: **while** not converged **do**
3:    **for** all $u \in U$ **do**
4:       **for** all $k \in \{1, 2, \ldots, K\}$ **do**
5:          Get the modified modified graphs $G_k$ by Eq. (9);
6:          Infer the causal environment label $z_{causal}$ from Eq. (43);
7:          Obtain the damage representation $\mathbf{x}_K^t$ by Eq. (14);
8:          // Forward process
9:          Sample $\mathbf{x}_k^{t-1}$ by feeding $z_{causal}$ and $\mathbf{x}_k^t$ into $f_{\theta_2}(z_{causal}, \mathbf{x}_k^t)$;
10:          // Reverse process
11:          Calculate $f_{\theta_2}(\mathbf{x}_k^{t-1})$ via Eq. (17) ;
12:          Calculate the gradients w.r.t. the loss in Eq. (27);
13:       **end for**
14:    **end for**
15:    Average the gradients over $|U|$ users and $K$ environments;
16:    Update $\omega$, $\theta_1$, and $\theta_2$ via AdamW optimizer;
17: **end while**
18: **Output:** $g_\omega(\cdot)$, $P_{\theta_1}(\cdot)$, and $f_{\theta_2}(\cdot)$.

- **LightGCN** [9]. It is an effective collaborative filtering method based on graph convolutional networks (GCNs) that streamline NGCF's message propagation scheme by eliminating non-linear projection and activation.
- **SGL** [40]. It uses LightGCN as the backbone and incorporates a series of structural augmentations to enhance representation learning.
- **SimGCL**[49]. This model employs a simple contrastive learning (CL) approach that avoids graph augmentations and introduces uniform noise into the embedding space to generate contrastive views.
- **LigthGCL** [1]. This uses LightGCN as its backbone and introduces uniform noise into the embedding space for contrastive learning without relying on graph augmentations
- **CDR** [32]. It captures preference shifts using a temporal variational autoencoder and learns the sparse influence from multiple environments.
- **InvPref** [39]. It is a general debiasing model that iteratively decomposes invariant and variant preferences from biased observational user behaviors by estimating heterogeneous environments corresponding to different types of latent bias.
- **InvCF** [54]. This model aims to mitigate popularity shift to discover disentangled representations that faithfully reveal the latent preference and popularity semantics without making any assumption about the popularity distribution.
- **AdvInfoNCE** [53]. It is an InfoNCE variant that leverages a detailed hardness-aware ranking criterion to improve the recommender's ability to generalize

- **AdvDrop** [52]. It is designed to alleviate general biases and inherent bias amplification in graph-based collaborative filtering by enforcing embedding-level invariance from learned bias-related views.
- **DR-GNN** [28]. A GNN-based OOD recommendation algorithm solves the data distribution shift via the Distributionally Robust Optimization theory.

## F Model Complexity Analysis

We analyze the time complexity of our CausalDiffRec across its different components: i) The environment generator has a time complexity of $O(k \cdot |E|)$, where $|E|$ is the size of the edge set and $k$ is the number of generated environments. ii) The most time-consuming part of the causal diffusion component is the diffusion generation. Initially, a Variational Graph Auto-Encoder (VGAE) maps the graph-structured data to a fixed distribution in latent space with a complexity of $O(L \cdot |E| \cdot d)$, where $L$ is the number of layers in the encoder part of VGAE, and $d$ represents the hidden layer dimensions. The overall time complexity of the diffusion model is $O(T \cdot L \cdot d)$, where $T$ is the number of time steps. iii) The environment inference component has a time complexity of $O(1)$. In summary, CausalDiffRec can achieve overall complexity comparable to state-of-the-art diffusion-based recommendation models such as DiffRec [33] and RecDiff [22].

