# OpenReview forum: "Graph Representation Learning via Causal Diffusion for Out-of-Distribution Recommendation"
_ACM.org/TheWebConf/2025/Conference — WWW 2025 Oral_

### Official Review · Reviewer_1rgP · 2024-11-25

**Novelty:** 4
**Technical Quality:** 4

**Review:**

Summary

This paper proposes a novel approach, graph representation learning via causal diffusion (CausalDiffRec), to address the challenge of out-of-distribution (OOD) recommendation in graph neural network (GNN)-based recommender systems. The authors construct a Structural Causal Model (SCM) to analyze the impact of environmental confounders on the generalization of GNN-based models, revealing that these confounders lead to unstable correlations and poor performance on OOD data. To tackle this issue, CausalDiffRec uses backdoor adjustment and variational inference to infer the real environmental distribution, eliminating the impact of confounders and learning invariant graph representations. The authors provide a theoretical derivation showing that optimizing the CausalDiffRec objective function encourages the model to learn environment-invariant representations, thereby achieving excellent generalization on OOD data. Extensive experiments validate the effectiveness of CausalDiffRec, with significant improvements over baseline methods on various datasets.

Strengths:

S1. The method is novel and technically sound. The proposed CausalDiffRec approach uses backdoor adjustment and variational inference to infer the real environmental distribution, eliminating the impact of confounders and learning invariant graph representations, thereby improving generalization on OOD data.

S2. The paper is well-written and easy to follow. The figures are clear and easy to understand.

S3. The paper provides a theoretical derivation showing that optimizing the CausalDiffRec objective function encourages the model to learn environment-invariant representations, achieving excellent generalization performance in recommendations under distribution shifts.

Weaknesses:

W1: Will $z_{causal}$ denotes the latent embeddings or latent distributions in line 542? Should we utilize the reparameterized method in finding $z_{causal}$?

W2: The authors should specifically point out what is the exact "environment" means in the paper.

W3: What does the green and gray dots represent in the Fig.2?

W4: How about the time and space complexity of your proposed method? The inference process of DDPM may be time-consuming.

**Questions:**

Please refer to the weaknesses above.

**Reviewer Confidence:**

4: The reviewer is certain that the evaluation is correct and very familiar with the relevant literature

**Scope:**

4: The work is relevant to the Web and to the track, and is of broad interest to the community

---

### Official Review · Reviewer_8Mrh · 2024-12-02

**Novelty:** 6
**Technical Quality:** 6

**Review:**

The paper "Graph Representation Learning via Causal Diffusion for Out-of-Distribution Recommendation" introduces a novel approach to address the Out-of-Distribution (OOD) recommendation problem by leveraging causal diffusion in graph representation learning.

The manuscript is well-structured and clearly articulates the research problem, methodology, and experimental results. The authors provide a comprehensive review of related works, effectively positioning their contribution within the existing literature. The methodology is detailed, and the experimental setup is robust, enhancing the reproducibility of the study.

The integration of causal diffusion into graph representation learning for OOD recommendation is a novel contribution. This approach distinguishes itself from traditional methods by incorporating causal inference to handle distribution shifts, offering a fresh perspective on the OOD recommendation challenge.

Addressing the OOD recommendation problem is crucial for the practical deployment of recommender systems in dynamic environments. The proposed method demonstrates significant improvements over baseline models, indicating its potential impact on real-world applications. The authors' approach not only advances theoretical understanding but also provides practical solutions for enhancing recommendation system performance in OOD scenarios.

Pros

1. The use of causal diffusion in graph representation learning for OOD recommendation is a novel and promising direction.
2.The paper includes extensive experiments across multiple real-world datasets, providing a thorough evaluation of the proposed method.
3.The methodology and results are presented clearly, facilitating understanding and reproducibility.

Cons
1. The proposed method may introduce additional computational overhead due to the complexity of causal diffusion processes.
2. While the method performs well on the datasets used, its scalability to larger, more complex networks remains to be fully assessed.

**Questions:**

1. How does the proposed method perform on larger datasets or networks with more complex structures? Are there any plans to address potential scalability issues?
2. While the paper compares the proposed method with baseline models, how does it perform relative to other recent approaches in OOD recommendation that also incorporate causal inference?
3. Does the causal diffusion process provide any insights into the underlying mechanisms of the recommendation system? How interpretable are the learned representations?

**Reviewer Confidence:**

3: The reviewer is confident but not certain that the evaluation is correct

**Scope:**

4: The work is relevant to the Web and to the track, and is of broad interest to the community

---

### Official Review · Reviewer_Gm2P · 2024-12-02

**Novelty:** 4
**Technical Quality:** 5

**Review:**

Graph Neural Network (GNN)-based recommendation models often assume independent and identically distributed (IID) data, but this fails with out-of-distribution (OOD) data, leading to poor performance. This study introduces **CausalDiffRec**, a method to improve OOD generalization by addressing environmental confounders (e.g., COVID-19) that disrupt correlations. Using a Structural Causal Model (SCM), it employs backdoor adjustment and the diffusion model to estimate true environmental distributions, guiding the learning of invariant graph representations. This process ensures robust recommendations despite distribution shifts. Theoretical analysis confirms the approach enhances generalization, with experiments showing significant improvements on multiple datasets.



Pros:

1. This paper addresses an urgent problem in recommender systems: trying to address distribution shifts when the cause of distribution shift is hidden.
2. The method in this paper is clearly illustrated and the authors provide theoretical results for their method.
3. The experiments are sufficient to support the superiority of the proposed method CausalDiffRec.

Cons:

1. The motivation why the authors use diffusion model for representation learning is unclear. In the Introduction, the authors use a motivation example to stress the urgency of designing methods to address distribution shifts (due to the underlying environments), thus motivating the use of invariant learning. However, why the authors use diffusion models needs to be clarified.
2. The authors use the invariant learning to address the distributional shift due to environments and claim it as a major difference from existing GNN-based recommendation methods in Sec 5.3. However, in Sec 3 (Methodology), I didn't see how the invariant learning is adopted (or I missed it).
3. Why there is no arraw from I to G in the causal graph? In addition, what is the difference between G and Y?

**Questions:**

1. The method uses an environment generator (line 480) to simulate multiple environments. Though the method is from literature, I wonder if this simulation process plays a role. Is it sufficient to simulate complex enough environments? Or if the actual distributional shift scenarios cannot be reflected by the environment generator, can the proposed module still work well for the ultimate goal? Maybe the authors can conduct experiments on more complex distributional shifts such as combining all three shifts.

2. In line 574, "sample the final embeddings for user preference modeling" is unclear to me.

3. In line 455, the proposed method "takes the user’s original interaction graph $G$ as input and generates a set of $K$ interaction graphs". However, the original interaction is regarded as an  interaction under a mixed of environments. I think the goal of this step is to simulate multiple environments of the interaction under each underlying environment. How to achieve the goal  just by  "taking the user’s original interaction graph $G$ as input and generates a set of $K$ interaction graphs"  is unclear.

4. The results of the effects of the number of environments $K$ are strange. The ideal pattern is that when the number of environments is lower than a specific value, the performance arises with the number of environments increasing and falls when the number of environments is higher than a specific value. However, in both the Food and KuaiRec datasets, the observed patterns are different from the ideal pattern. The authors should explain more about it.

**Reviewer Confidence:**

3: The reviewer is confident but not certain that the evaluation is correct

**Scope:**

3: The work is somewhat relevant to the Web and to the track, and is of narrow interest to a sub-community

---

### Official Review · Reviewer_TrdZ · 2024-12-02

**Novelty:** 5
**Technical Quality:** 5

**Review:**

This works studied the ood problem in graph-based recsys, and proposed to alleviate such issue via caudal diffusion. The authors carefully designed a causal model for the problem, with the DAG well defined. The experiments also showed good performances.

S1: the problem is interestingly studied with a well-defined causal model as the design motivation of the proposed method.

S2: The experiments showcased consistent improvements, with ablative studies.

W1: my main concern is on the efficiency and scalability, which the authors didn't discuss.

**Questions:**

The proposed method seems to be adding quite some overhead on top of the existing methods. How efficient or scalable is the proposed method? I'd appreciate some analysis as well as empirical comparisons on these.

**Reviewer Confidence:**

2: The reviewer is willing to defend the evaluation, but it is likely that the reviewer did not understand parts of the paper

**Scope:**

3: The work is somewhat relevant to the Web and to the track, and is of narrow interest to a sub-community